# Mortality Risk Factors in Patients Admitted with the Primary Diagnosis of Tracheostomy Complications: An Analysis of 8026 Patients

**DOI:** 10.3390/ijerph19159031

**Published:** 2022-07-25

**Authors:** Lior Levy, Abbas Smiley, Rifat Latifi

**Affiliations:** 1School of Medicine, New York Medical College, Valhalla, NY 10595, USA; llevy6@student.nymc.edu (L.L.); abbas.smiley@wmchealth.org (A.S.); 2Westchester Medical Center, New York Medical College, Valhalla, NY 10595, USA; 3Department of Surgery, University of Arizona, Tucson, AZ 85721, USA; 4Ministry of Health, 10000 Pristina, Kosovo

**Keywords:** tracheostomy, in-hospital mortality, hospital length of stay, emergency admission

## Abstract

Background: Tracheostomy is a procedure commonly conducted in patients undergoing emergency admission and requires prolonged mechanical ventilation. In the present study, the aim was to determine the prevalence and risk factors of mortality among emergently admitted patients with tracheostomy complications, during the years 2005–2014. Methods: This was a retrospective cohort study. Demographics and clinical data were obtained from the National Inpatient Sample, 2005–2014, to evaluate elderly (65+ years) and non-elderly adult patients (18–64 years) with tracheostomy complications (ICD-9 code, 519) who underwent emergency admission. A multivariable logistic regression model with backward elimination was used to identify the association between predictors and in-hospital mortality. Results: A total of 4711 non-elderly and 3315 elderly patients were included. Females included 44.5% of the non-elderly patients and 47.6% of the elderly patients. In total, 181 (3.8%) non-elderly patients died, of which 48.1% were female, and 163 (4.9%) elderly patients died, of which 48.5% were female. The mean (SD) age of the non-elderly patients was 50 years and for elderly patients was 74 years. The mean age at the time of death of non-elderly patients was 53 years and for elderly patients was 75 years. The odds ratio (95% confidence interval, *p*-value) of some of the pertinent risk factors for mortality showed by the final regression model were older age (OR = 1.007, 95% CI: 1.001–1.013, *p* < 0.02), longer hospital length of stay (OR = 1.008, 95% CI: 1.001–1.016, *p* < 0.18), cardiac disease (OR = 3.21, 95% CI: 2.48–4.15, *p* < 0.001), and liver disease (OR = 2.61, 95% CI: 1.73–3.93, *p* < 0.001). Conclusion: Age, hospital length of stay, and several comorbidities have been shown to be significant risk factors in in-hospital mortality in patients admitted emergently with the primary diagnosis of tracheostomy complications. Each year of age increased the risk of mortality by 0.7% and each additional day in the hospital increased it by 0.8%.

## 1. Introduction

Tracheostomy is one of the most frequently performed procedures in intensive care medicine [1] aiming to facilitate the recovery of critically ill patients via liberating them from mechanical ventilation, increasing the ability to mobilize patients with chronic respiratory failure, and reducing the dosage of sedation [2]. Every year, approximately 800,000 U.S. residents need invasive mechanical ventilation [3] accounting for at least 25% of intensive care admissions in many hospitals [4]. Tracheostomy use increased rapidly through 2008, at which point we began to observe yearly declines [5]. In recent years, the number of chronically critically ill patients requiring prolonged mechanical ventilation (MV) and receiving a tracheostomy is steadily increasing [6]. Tracheostomy may improve aspects of care for MV patients [5], such as sedation requirements, mobility, and oral feeding [7,8].

Complication rates currently exceed 50% and can result in life threats such as decannulation, obstruction, and hemorrhage [9]. The use of tracheostomies has substantial implications on cost, resource use, and outcomes during acute hospitalization and following hospital discharge [10,11,12]. Insufficient emphasis had been placed on drawing a clear personalized patient profile and developing practices for the management and understanding of the individual clinical, socioeconomic, and demographic differences of patients undergoing emergency surgery for tracheostomy complications. These personal data are essential for clinical decision-making through early initiation of treatment in high-risk patients, and therefore is a determinant of a successful outcome of the treatment. The use of artificial intelligence through machine-learning classifiers was suggested for the early identification of patients at risk for prolonged mechanical ventilation and tracheostomy. The application of these identification techniques could lead to improved outcomes by allowing for early intervention [13]. This would require elucidating the pertinent risk factors of mortality, complications, and a longer hospital length of stay.

The variation in demographic, clinical, and hospitalization characteristics of the patients requires a thorough analysis of clinical data, management implementation, and treatment outcomes [14]. It is incumbent for the emergency team to be able to promptly identify the signs of tracheostomy-related emergencies, issue preventive strategies, allocate resources, and provide stabilization. The clinical relevance of these complications is crucial, as their exemplifications range from barely symptomatic to failure to wean from the mechanical ventilator to life-threatening hemorrhage. Sound knowledge of tracheostomy risks and the increasing rate of complications is crucial for the performing surgeon and the supporting health care team. As such, the aim of this study is to evaluate the specific risk factors for mortality in patients admitted emergently with the primary diagnosis of tracheostomy.

## 2. Materials and Methods

The Healthcare Cost and Utilization Project (HCUP) was instituted to produce multistate, administrative, and population-based statistics on patients in a systematic format. The data are directed toward health services research to reinforce healthcare provision. The National Inpatient Sample (NIS), a broad administrative database formed by the Agency for Healthcare Research and Quality (AHRQ), has been progressively employed as a country-wide public data source and carries much potential and supports the assessment of patterns of care and outcomes for research. It enables an innovative perspective to examine disease conditions, optimal care, and patient outcomes. The NIS applies the process of weighting when generating the sample of discharges from community hospitals in the US, excluding rehabilitation centers and long-term acute care facilities. This approach of stratification makes it feasible to produce a national estimate of hospitalizations for attainable factors. This retrospective cohort study extracted data on adult and elderly patients with tracheostomy complications that had emergency general surgery (EGS) procedures. The sample was pulled from the NIS 2005–2014. The ICD-9 code to identify patients with tracheostomy complications was 519. Table 1 summarizes the ICD-9 codes for the operations and invasive diagnostic procedures on both the digestive and respiratory systems. The following characteristics of patients and hospitals were collected and analyzed: Age, gender, race, income quartile, hospital location (rural, urban–non-teaching, urban–teaching), healthcare insurance (Medicare, Medi-caid, private insurance, self-paid, and no charge), tracheostomy complication (infection, mechanical complication, unspecified, other), required an invasive diagnostic procedure vs. not, required surgical procedure vs. not, days to the first procedure, length of hospital stay, and total charges. Post-procedure mortality was limited to immediate hospitalization, not including future clinical courses in referral wards or institutions. R software was used for the statistical analysis, and *p* < 0.01 was set as significant.

### Statistical Analysis

Descriptive and analytical statistical indicators were used to present the findings. The mean, standard deviation (SD), and confidence interval at 95% (CI) were calculated for numerical variables. The comparisons used a chi-square test for categorical variables and a *t*-test for continuous variables. The data were compared in three different ways. The first way examined gender within the two age categories. The second compared the surviving vs. deceased patients for both adult and elderly patients. Finally, we looked at adult and elderly patients who either underwent an operation or did not. The ability of different variables to predict mortality was evaluated by a backward multivariable logistic regression analysis. *P*-values less than 0.05 were considered significant. All analyses were performed using SPSS software version 26 (SPSS Inc., Chicago, IL, USA) and R statistical software (Foundation for Statistical Computing, Vienna, Austria).

## 3. Results

### 3.1. Gender Differences

#### 3.1.1. Non-Elderly Patients

The mean age of the 181 patients who passed away during the study period was 53 years, of which 94 were males (52%) and 87 were females (48%). Most patients were white, of income quartile 1, funded mostly by Medicaid, and were admitted to a teaching hospital. The main comorbidities were hypertension, deficiency anemias, congestive heart failure, uncomplicated diabetes, obesity, and renal failure. Males had substantially greater rates of alcohol abuse, metastatic cancer, and solid tumor as females manifested higher rates of congestive heart failure, chronic pulmonary disease, uncomplicated diabetes, hypothyroidism, depression, and obesity. No significant difference was shown between genders in terms of the invasive diagnostic procedure rate, surgical procedure rate, or mortality rate. Patients’ characteristics and clinical data are summarized in Table 2.

#### 3.1.2. Elderly Patients

Most patients were white, of income quartile 1, funded mainly by Medicare, and were admitted to an urban-teaching hospital. Some of the most common comorbidities were hypertension, congestive heart failure, uncomplicated diabetes, and fluid/electrolyte disorders. Males demonstrated a substantially greater rate of alcohol abuse, metastatic cancer, and solid tumors and females manifested a significantly higher rate of congestive heart disease, depression, uncomplicated diabetes, hypothyroidism, pulmonary circulation disorders, and obesity. Women also had a higher surgical procedure rate and a longer hospital length of stay (HLOS). No significant discrepancy was recognized between genders in terms of the invasive diagnostic procedure rate or mortality rate. Patients’ characteristics and clinical data are summarized in Table 2.

### 3.2. Mortality

#### 3.2.1. Non-Elderly Patients

In total, 96.2% survived and 3.8% died. The mean (SD) age of the patients who survived was significantly lower than the patients who died, at 50 vs. 54 years, respectively. Of the deceased patients, 94 were males (52%) and 87 were females (48%), with a similar mean age. When comparing the deceased to the surviving patients, significant differences were identified in the comorbidities. The deceased manifested significantly greater rates of coagulopathy, liver disease, fluid/electrolyte disorders, metastatic cancer, and renal failure. Furthermore, the deceased had significantly higher rates of tracheostomy complications, which were not mechanical nor infections. Findings are summarized in Table 3. Most digestive tract surgical procedures were operations on the stomach (ICD-9 codes 43.0–44.03, 44.21–44.99) at 56.0%, which included gastrotomy and vagotomy. Furthermore, 65.3% of digestive tract invasive diagnostic procedures were those on the intestines (ICD-9 codes 45.11–45.29) such as endoscopy and biopsy. In total, 88.2% of surgical procedures on the respiratory system (ICD-9 30.01–31.3, 31.5–31.99) were operations on the larynx and trachea. However, when it comes to invasive diagnostic procedures, 63.0% were performed on the lungs and bronchus (ICD-9 codes 33.20–33.29).

#### 3.2.2. Elderly Patients

In total, 95.1% survived and 4.9% passed away (Table 3). In the surviving group, 1495 were females (47.5%) and 1650 were males (52.5%). In the deceased group, 79 were females (48.5%) and 84 were males (51.5%). When comparing the deceased to the surviving patients, differences were recognized in several comorbidities. Most elderly patients with tracheostomy complications were Caucasian males. The main comorbidities were hypertension, uncomplicated diabetes, and fluid/electrolyte disorders. The deceased had significantly greater rates of comorbidities with fluid/electrolyte disorders and coagulopathy. Patients’ characteristics and clinical data are summarized in Table 3. In total, 63.8% of the mentioned surgical procedures on the gastrointestinal system in this patient group were performed on the stomach, which includes operations such as gastrotomies. On the other hand, 70.3% of the invasive diagnostic procedures on the gastrointestinal tract were performed on the intestines, which includes biopsies and endoscopies. Looking at the respiratory system, 85.1% of the surgical procedures were performed on the larynx and trachea, for example, operations such as laryngectomies and tracheostomies. Moreover, 62.9% of the respiratory system’s invasive diagnostic procedures were performed on the lung and bronchus in this population, which include tissue biopsies and bronchoscopies.

### 3.3. Operation vs. Non-Operation

#### 3.3.1. Non-Elderly Patients

The stratified analysis, based on the surgical procedure status, is presented in Table 4. In total, 1348 (28.6%) had a surgical procedure. In both groups, most patients were males. The mean (SD) age in the surgical group was significantly lower in comparison to the non-operated group. The racial breakdown by the proportion of cases in decreasing order was White, Black, Hispanic, Asian/Pacific Islander, and Native American. Most patients were of income quartile 1, funded mainly by Medicaid, and were admitted to urban teaching hospitals. In the group that had a surgical procedure, the rate of solid tumors was significantly lower in comparison to the other group. They had a substantially greater rate of mechanical complications and a significantly lower rate of other categories of tracheostomy complications in comparison to the non-operated group, a significantly greater rate of respiratory system invasive diagnostic procedures, a higher rate of digestive system invasive diagnostic procedures, as well as a longer time to invasive diagnostic procedures and longer HLOS.

#### 3.3.2. Elderly Patients

The stratified analysis, based on the surgery status, is presented in Table 4. In total, 764 (23.0%) had surgery while 2551 (77.0%) did not. The racial breakdown by the proportion of cases in decreasing order was White, Black, Hispanic, Asian/Pacific Islander, and Native American. Most patients were of income quartile 1 and were admitted to urban teaching hospitals. The operated group demonstrated a substantially higher rate of mechanical complications and a significantly lower rate of other categories of tracheostomy complications compared to the non-operated group, a substantially greater rate of respiratory system invasive diagnostic procedure, a higher rate of digestive system invasive diagnostic procedure, as well as a longer time to an invasive diagnostic procedure and a longer HLOS.

### 3.4. Risk Factors of Mortality

The findings of the multivariable backward logistic regression model for risk factors of mortality are presented in Table 5. Common variables included in the model were age, gender, social factors, lifestyle elements, and comorbidities. Age, HLOS, and several comorbidities emerged as significant risk factors for in-hospital mortality in patients admitted emergently with the primary diagnosis of tracheostomy complications. In patients with tracheostomy complications, each additional year of age was associated with a 0.7% increase in the odds of mortality. Each additional day in the hospital raised the odds of mortality by 0.8%. Cardiac disease increased the odds of dying by 3.21-fold, while liver disease raised it by 2.61-fold (Table 5).

### 3.5. Lifestyle, Comorbidities, and Secondary Diagnoses

Table 6 summarizes the lifestyle, comorbidities, and secondary diagnoses of patients. There was no significant difference in lifestyle elements between the surviving and deceased patients. Almost the same comorbidities that stayed in the final regression model as the significant predictors of mortality were significantly more prevalent in deceased patients than in surviving ones.

## 4. Discussion

The primary aim of this study was to evaluate associations between demographics, socioeconomic status, surgical status, comorbidities, and HLOS with overall mortality in emergently admitted patients with the primary diagnosis of tracheostomy complications. Older age and increased hospital length of stay were among the most important significant risk factors for mortality.

### 4.1. The Impact of HLOS on Mortality

Longer hospital stay was associated with adverse outcomes. Prolonged HLOS for older patients in the emergency department has been shown to be associated with a higher risk of hospitalization and adverse outcomes [15]. Mowery et al. collected evidence that the emergency department length of stay is an independent predictor of hospital mortality in trauma activation patients [16]. Bohm et al. showed that shorter time to surgery and decreased HLOS improved in-hospital and 1-year mortality [17]. Furthermore, Zhang added that prolonged EDLOS is independently associated with an increased risk of hospital mortality in patients with sepsis requiring ICU admission [18].

### 4.2. The Impact of Gender on Mortality

In agreement with our results, Mehta et al. [5] have shown that across all study years, the likelihood of tracheostomy complications was significantly affected by gender and was highly prevalent in males. However, our findings showed that mortality was not affected by gender.

### 4.3. The Impact of Age on Mortality

Additionally, we have demonstrated that age is a major risk as well, correlating with the increase in comorbidities, the increased rate of complications, and an increased mortality rate in non-elderly. The older the patient, the lower the likelihood of survival. These results are supported by Tamir et al. who have shown that comorbid conditions and subject age had a greater association with the 30-day mortality rate [19].

Overall, mortality associated with tracheostomy complications was relatively low at 3.8% for adults and 4.9% in elderly patients, but this is higher than that of Kligerman et al.’s study who showed 1.4% in a cohort of 38,293 patients with the primary diagnosis of tracheostomy complication [20].

### 4.4. The Impact of Comborbidities on Mortality

Our results have shown that comorbidities such as coagulopathy, fluid/electrolyte disorders, and metastatic cancer were significant predictors of mortality in non-elderly patients, and fluid/electrolyte disorders were the single significant predictor of mortality in elderly patients. In contrast, Kligerman et al. manifested that tracheostomy infection was the most significant predictor of mortality [20].

In the design of these surveys, considerable effort was dedicated to attaining a representative sample to eliminate biases that may have been present in smaller or regional surveys. The management of emergent tracheostomy complications may be further complicated by the underlying condition of the patient who originally necessitated the placement of the tracheostomy. Among elderly and non-elderly primary diagnosis patients, tracheostomy complications were shown to be comorbid in males, chiefly with alcohol abuse, solid tumors, and metastatic cancer, while in females, they were comorbid regarding congestive heart failure, chronic pulmonary disease, uncomplicated diabetes, obesity, and hypothyroidism. Numerous studies have manifested the adverse impact that obesity, anemia, hypertension, diabetes, and cardiovascular disease have on tracheostomy patients’ conditions. Diabetes mellitus is one of the most prevalent conditions in the elderly and is associated with increased susceptibility to infectious diseases, and considerable morbidity and mortality, mainly from cardiovascular and renal complications [21]. Kumarasinghe et al. manifested that diabetes is associated with an increase in colonization and infection of tracheostomy tubes [22].

Stomal stenosis develops secondarily to bacterial infection, which conspires to weaken the anterior and lateral tracheal walls. Stomal granulation tissue frequently develops, and nearly all patients have some degree of tracheal narrowing at the site of the tracheostoma [23].

A high percentage of prolonged mechanically ventilated patients showed anemia on admission [24]. Retrospective studies of critically ill patients showed a positive correlation between transfusions with prolonged mechanical ventilation, increased mortality rates, and increased risk of nosocomial infections, which in turn could adversely affect weaning outcomes [25]. The tracheostomy-related complication rate is significantly higher for obese patients [26], especially with a body mass index ≥ 35 and especially in the intraoperative and early postoperative time periods [27]. Obesity was found to be independently associated with an increased risk of overall complications, developing acute renal failure, and having unplanned 30-day readmission following tracheostomy [28]. Obese patients have a greater likelihood of complications and an increased risk of remaining tracheostomy-dependent [29]. Although open tracheostomy in morbidly obese patients is increasing in demand, the procedure can be predictably performed albeit at a much longer duration and a higher perioperative complication rate compared with the traditional tracheostomy [30]. Studies showed that hypertensive patients have an exaggerated hemodynamic stress response [31]. In critically ill patients, this stress response might result in slower recovery and an increased risk of mortality. Studies have shown that bleeding is the most common cause of morbidity and mortality after tracheostomy. However, in the ED setting, some bedridden cardiovascular patients are on full anticoagulation to prevent thromboembolic events in bedridden patients [32,33]. The management of a preoperative cancer patient should consider that functional patient assessment and pulmonary function testing are key to preoperative assessment. By optimizing the patients’ condition ahead of the tracheostomy, the risk of complications is limited.

### 4.5. The Impact of an Invasive Diagnostic Procedure on Complications on Mortality

Our results demonstrate that white male patients, non-elderly as well as elderly, who had an invasive diagnostic procedure had a high positive correlation with mechanical complications, infection, and other complications, as well as a higher rate of respiratory or digestive systems surgical procedures and significantly prolonged HLOS. Invasive diagnostic procedures are complementary to serologic and non-invasive studies and assist in rapidly establishing an accurate diagnosis, which allows the initiation of appropriate therapy and may improve outcomes with relative safety [34]. Halliday et al. have presented similar results showing elevated complication rates in a study of invasive diagnostic procedures for lung abnormalities [35]. Evaluating patients based on invasive diagnostic procedure outcomes enables forming a management plan for surgical vs. non-surgical treatment.

### 4.6. Strengths of the Study

The primary strengths of this study relate to the large, nationally representative patient population across a wide spectrum of hospitals and geographic locations. This analysis was recorded in an exhaustive nationwide distinctive database during a 10-year period in the United States during 2005–2014. The large patient population enabled us to identify the predictors of mortality and increased HLOS associated with tracheostomy complications. Thus, our results are likely to be generalizable across a range of locations and practice settings. The sample size was large enough for accurate analysis with each statistical method. Most previous studies assessing the prevalence of tracheostomy complications were confined to small populations from single hospitals or geographic regions. Given the numerous gaps in our understanding of tracheostomy complications, the time has come for research to be designed to expand the evidence base. Our study provides avenues for future investigations. Understanding potential trajectories in morbidity and mortality is crucial to guiding long-term investments and policy implementation.

### 4.7. Limitations of the Study

Analysing the data, several limitations are noted in this study. First, future surveys should aim to include the exact type of tracheostomy used such as surgical tracheostomy, percutaneous dilatational tracheotomy, ultrasound-guided percutaneous tracheostomy, conventional percutaneous tracheostomy, or bronchoscopic guided percutaneous tracheostomy. Lately, ultrasound-guided percutaneous tracheostomy (USPCT) has become a routine practice in ED with evident advantages [36]. It facilitates the clinician to identify the vascular structures and thyroid, delineate the airway [37], evaluate the thickness of the skin over the neck, and visualize the needle and guide wire passage [38]. A percutaneous approach offers fewer surgical-site infections and postsurgical bleeding than a surgical approach [39]. A surgical placement, on the other hand, possesses a lower risk of injury to the posterior tracheal wall [39]. Furthermore, the timing of the tracheostomy appears to be crucial [40]. Early’ and ‘late’ tracheostomies are two categories of the timing of tracheostomy. Evidence on the advantages attributed to early vs. late tracheostomy shows it reduces sedative use, allows early oral feeding [41], early mobility [42], and improves physiology [43], decreases the duration of mechanical ventilation and ICU stay, shortens hospital stays, and lowers mortality rates [44]. Additionally, indicating the exact specification data of the surgeon’s years of experience is essential. Otorhinolaryngology-Head and Neck Surgeons (ORL-HNS), for example, are familiar with the anatomy of upper airways, which is an important underlying factor in reducing the incidence of possible complications [26]. Furthermore, the severity of comorbidity, the type of anesthesia (local or general), the type of tracheal incision, and the exact cause of death should be specified for future analysis as well. Moreover, it is advised to classify the complication severity according to the Clavien Dindo classification [45]. Differentiative analysis including all these factors will shed light and guide future practice management of emergency departments.

## 5. Conclusions

Little is known about population-based trends in the use of a tracheostomy for mechanical ventilation in the United States. The results of the present study may be instrumental in reducing the risk of complications after a tracheostomy and decreasing HLOS and its associated mortality. The risk factors for mortality were advanced age, increased HLOS, cardiac disease, and liver disease. Some of the causes of death included tuberculosis, coagulopathy, cardiac disease, and tobacco use, among many others. Identifying system factors and standardizing care among specialties will help guide management when patients arrive in the emergency department.

## Figures and Tables

**Table 1 ijerph-19-09031-t001:** Procedures of emergently admitted patients with the primary diagnosis of tracheostomy complications.

Operations on the Digestive System (ICD 9)
Operations on Esophagus (42.01–42.19, 42.31–42.99)
Operations on Stomach (43.0–44.03, 44.21–44.99)
Operations on Intestine (45.00–45.03, 45.30–46.99)
Operations on Appendix (47.01–47.99)
Operations on Rectum, Rectosigmoid, and Perirectal Tissue (48.0–48.1, 48.31–48.99)
Operations on Anus (49.01–49.12, 49.31–49.99)
Operations on Liver (50.0, 50.21–50.99)
Operations on Gallbladder and Biliary Tract (51.01–51.04, 51.21–51.99)
Operations on Pancreas (52.01–52.09, 52.21–52.99)
Operations on Hernia (53.00–53.9)
Operations on Other Operations on Abdominal Region (54.0–54.19, 54.3–54.99)
**Invasive Diagnostic Procedures on the Digestive System (ICD 9)**
Invasive Diagnostic Procedure on Esophagus (42.21–42.29)
Invasive Diagnostic Procedure on Stomach (44.11–44.19)
Invasive Diagnostic Procedure on Intestine (45.11–45.29)
Invasive Diagnostic Procedure on Rectum, Rectosigmoid, and Perirectal Tissue (48.21–48.29)
Invasive Diagnostic Procedure on Anus (49.21–49.29)
Invasive Diagnostic Procedure on Liver (50.11–50.19)
Invasive Diagnostic Procedure on Gallbladder and Biliary Tract (51.10–51.19)
Invasive Diagnostic Procedure on Pancreas (52.11–52.19)
Invasive Diagnostic Procedure on Other Operations on Abdominal Region (54.21–54.29)
**Operations on the Respiratory System (ICD 9)**
Operations on Larynx and Trachea (30.01–31.3, 31.5–31.99)
Operations on Lung and Bronchus (32.01–33.1, 33.31–33.99)
Operations on Chest Wall, Pleura, Mediastinum, and Diaphragm (34.01–34.1, 34.3–34.99)
**Invasive Diagnostic Procedures on the Respiratory System (ICD 9)**
Invasive Diagnostic Procedure on Larynx and Trachea (31.41–31.49)
Invasive Diagnostic Procedure on Lung and Bronchus (33.20–33.29)
Invasive Diagnostic Procedure on Chest Wall, Pleura, Mediastinum, and Diaphragm (34.20–34.29)

**Table 2 ijerph-19-09031-t002:** Characteristics of emergently admitted patients with the primary diagnosis of tracheostomy complications (NIS 2005–2014). Data were stratified according to gender categories. * *p* < 0.05.

Patient Characteristics		Adults (18–64), N (%)		Elderly (65+), N (%)	
		Male	Female	Male	Female
All Cases		2616 (55.5%) *	2095 (44.5%) *	1737 (52.4%) *	1578 (47.6%) *
Race	White	1278 (57.0%) *	914 (50.9%) *	1058 (70.4%) *	821 (60.9%) *
	Black	545 (24.3%) *	606 (33.7%) *	225 (15.0%) *	313 (23.2%) *
	Hispanic	271 (12.1%) *	176 (9.8%) *	121 (8.1%) *	126 (9.3%) *
	Asian/Pacific Islander	60 (2.7%) *	44 (2.4%) *	46 (3.1%) *	32 (2.4%) *
	Native American	10 (0.4%) *	10 (0.6%) *	5 (0.3%) *	7 (0.5%) *
	Other	77 (3.4%) *	47 (2.6%) *	48 (3.2%) *	49 (3.6%) *
Income Quartile	Quartile 1	894 (35.4%) *	823 (40.1%) *	477 (28.2%)	477 (30.9%)
	Quartile 2	630 (25.0%) *	491 (23.9%) *	426 (25.2%)	379 (24.5%)
	Quartile 3	603 (23.9%) *	422 (20.5%) *	425 (25.1%)	367 (23.8%)
	Quartile 4	396 (15.7%) *	318 (15.5%) *	365 (21.6%)	321 (20.8%)
Insurance	Private Insurance	640 (24.5%) *	470 (22.5%) *	159 (9.2%) *	98 (6.2%) *
	Medicare	845 (32.3%) *	713 (34.1%) *	1475 (85.0%) *	1395 (88.6%) *
	Medicaid	902 (34.5%) *	806 (38.5%) *	52 (3.0%) *	62 (3.9%) *
	Self-Pay	103 (3.9%) *	49 (2.3%) *	16 (0.9%) *	6 (0.4%) *
	No Charge	8 (0.3%) *	9 (0.4%) *	0 (0%) *	1 (0.1%) *
	Other	115 (4.4%) *	44 (2.1%) *	33 (1.9%) *	13 (0.8%) *
Hospital Location	Rural	142 (5.4%)	130 (6.2%)	100 (5.8%)	67 (4.2%)
	Urban: Non-Teaching	772 (29.5%)	585 (27.9%)	590 (34.0%)	540 (34.2%)
	Urban: Teaching	1702 (65.1%)	1380 (65.9%)	1047 (60.3%)	971 (61.5%)
Comorbitdities	AIDS	21 (0.8%)	14 (0.7%)	0 (0%)	0 (0%)
	Alcohol Abuse	155 (5.9%) *	40 (1.9%) *	57 (3.3%) *	10 (0.6%) *
	Deficiency Anemias	541 (20.7%) *	498 (23.8%) *	485 (27.9%)	463 (29.3%)
	Rheumatoid Arthritis	17 (0.6%) *	65 (3.1%) *	20 (1.2%) *	55 (3.5%) *
	Chronic Blood Loss	31 (1.2%)	18 (0.9%)	26 (1.5%)	24 (1.5%)
	Congestive Heart Failure	351 (13.4%) *	419 (20.0%) *	375 (21.6%) *	468 (29.7%) *
	Chronic Pulmonary Disease	884 (33.8%) *	910 (43.4%) *	828 (47.7%)	783 (49.6%)
	Coagulopathy	122 (4.7%)	75 (3.6%)	83 (4.8%)	58 (3.7%)
	Depression	287 (11.0%) *	355 (16.9%) *	133 (7.7%) *	200 (12.7%) *
	Diabetes, Uncomplicated	661 (25.3%) *	726 (34.7%) *	542 (31.2%) *	630 (39.9%) *
	Diabetes, Chronic Complications	76 (2.9%) *	114 (5.4%) *	58 (3.3%)	64 (4.1%)
	Drug Abuse	83 (3.2%)	70 (3.3%)	7 (0.4%)	7 (0.4%)
	Hypertension	1209 (46.2%) *	1131 (54.0%) *	1048 (60.3%) *	1030 (65.3%) *
	Hypothyroidism	249 (9.5%) *	338 (16.1%) *	283 (16.3%) *	380 (24.1%) *
	Liver Disease	103 (3.9%) *	48 (2.3%) *	31 (1.8%) *	15 (1.0%) *
	Lymphoma	13 (0.5%)	6 (0.3%)	11 (0.6%)	17 (1.1%)
	Fluid/Electrolyte Disorders	602 (23.0%)	529 (25.3%)	473 (27.2%) *	481 (30.5%) *
	Metastatic Cancer	175 (6.7%) *	74 (3.5%) *	124 (7.1%) *	53 (3.4%) *
	Other Neurological Disorders	454 (17.4%)	358 (17.1%)	235 (13.5%) *	253 (16.0%) *
	Obesity	422 (16.1%) *	600 (28.6%) *	122 (7.0%) *	247 (15.7%) *
	Paralysis	420 (16.1%) *	269 (12.8%) *	134 (7.7%)	122 (7.7%)
	Peripheral Vascular Disorders	75 (2.9%)	43 (2.1%)	129 (7.4%) *	88 (5.6%) *
	Psychoses	107 (4.1%) *	130 (6.2%) *	43 (2.5%)	56 (3.5%)
	Pulmonary Circulation Disorders	110 (4.2%) *	122 (5.8%) *	57 (3.3%) *	101 (6.4%) *
	Renal Failure	276 (10.6%) *	287 (13.7%) *	290 (16.7%)	276 (17.5%)
	Solid Tumor	353 (13.5%) *	143 (6.8%) *	294 (16.9%) *	99 (6.3%) *
	Peptic Ulcer	0 (0%)	0 (0%)	0 (0%)	0 (0%)
	Valvular Disease	56 (2.1%)	62 (3.0%)	95 (5.5%)	102 (6.5%)
	Weight Loss	237 (9.1%)	172 (8.2%)	178 (10.2%)	182 (11.5%)
Tracheostomy Complication	Unspecified Complication	23 (0.9%) *	23 (1.1%) *	17 (1.0%) *	15 (1.0%) *
	Infection	341 (13.0%) *	217 (10.4%) *	172 (9.9%) *	114 (7.2%) *
	Mechanical Compilation	802 (30.7%) *	774 (36.9%) *	469 (27.0%) *	540 (34.2%) *
	Other Tracheostomy Complication	1450 (55.4%) *	1081 (51.6%) *	1079 (62.1%) *	909 (57.6%) *
Invasive Diagnostic Procedure		979 (37.4%)	798 (38.1%)	640 (36.8%)	565 (35.8%)
Surgical Procedure		727 (27.8%)	621 (29.6%)	373 (21.5%) *	391 (24.8%) *
Deceased		94 (3.6%)	87 (4.2%)	84 (4.8%)	79 (5.0%)
		**Mean (SD)**	**Mean (SD)**	**Mean (SD)**	**Mean (SD)**
Age, Years		49.51 (12.32)	50.09 (11.42)	74.15 (6.86)	74.59 (7.00)
Time to Invasive Diagnostic Procedure, Days		1.72 (3.49) *	1.90 (3.15) *	1.85 (3.44)	1.92 (4.65)
Time to Surgical Procedure, Days		2.43 (4.12)	2.74 (4.61)	2.65 (3.84)	3.02 (5.12)
Hospital Length of Stay, Days		6.33 (9.95) *	6.97 (10.40) *	6.43 (8.43) *	6.70 (8.65) *
Total Charges, Dollars		50,206 (77,755) *	56,312 (91,144) *	50,359 (76,610) *	55,916 (101,092) *

**Table 3 ijerph-19-09031-t003:** Characteristics of emergently admitted patients with the primary diagnosis of tracheostomy complications. Data were classified according to outcome categories, NIS 2005–2014.

Patients’ Characteristics		Adult (18–64), N (%)			Elderly (65+), N (%)		
		Survived	Deceased	*p*	Survived	Deceased	*p*
All Cases		4524 (96.2%)	181 (3.8%)		3145 (95.1%)	163 (4.9%)	
Gender, Female		2006 (44.3%)	87 (48.1%)	0.320	1495 (47.5%)	79 (48.5%)	0.820
Race	White	2100 (54.1%)	90 (60.0%)	0.650	1788 (66.1%)	88 (62.4%)	0.500
	Black	1113 (28.7%)	38 (25.3%)		510 (18.8%)	27 (19.1%)	
	Hispanic	433 (11.1%)	12 (8.0%)		232 (8.6%)	15 (10.6%)	
	Asian/Pacific Islander	99 (2.5%)	5 (3.3%)		71 (2.6%)	7 (5.0%)	
	Native American	19 (0.5%)	1 (0.7%)		12 (0.4%)	0 (0%)	
	Other	120 (3.1%)	4 (2.7%)		93 (3.4%)	4 (2.8%)	
Income Quartile	Quartile 1	1662 (37.8%)	54 (31.4%)	0.380	900 (29.3%)	54 (34.2%)	0.130
	Quartile 2	1075 (24.4%)	44 (25.6%)		768 (25.0%)	35 (22.2%)	
	Quartile 3	980 (22.3%)	44 (25.6%)		761 (24.8%)	29 (18.4%)	
	Quartile 4	682 (15.5%)	30 (17.4%)		643 (20.9%)	40 (25.3%)	
Insurance	Private Insurance	1056 (23.4%)	52 (28.7%)	0.230	248 (7.9%)	9 (5.5%)	0.860
	Medicare	1493 (33.1%)	65 (35.9%)		2717 (86.5%)	147 (90.2%)	
	Medicaid	1653 (36.6%)	53 (29.3%)		109 (3.5%)	4 (2.5%)	
	Self-Pay	149 (3.3%)	3 (1.7%)		21 (0.7%)	1 (0.6%)	
	No Charge	16 (0.4%)	1 (0.6%)		1 (0.0%)	0 (0%)	
	Other	150 (3.3%)	7 (3.9%)		44 (1.4%)	2 (1.2%)	
Hospital Location	Rural	265 (5.9%)	7 (3.9%)	0.380	157 (5.0%)	10 (6.1%)	0.800
	Urban: Non-Teaching	1297 (28.7%)	58 (32.0%)		1071 (34.1%)	56 (34.4%)	
	Urban: Teaching	2962 (65.5%)	116 (64.1%)		1917 (61.0%)	97 (59.5%)	
Comorbidities	AIDS	35 (0.8%)	0 (0%)	0.640	0 (0%)	0 (0%)	
	Alcohol Abuse	189 (4.2%)	6 (3.3%)	0.570	66 (2.1%)	1 (0.6%)	0.260
	Deficiency Anemias	992 (21.9%)	45 (24.9%)	0.350	897 (28.5%)	47 (28.8%)	0.930
	Rheumatoid Arthritis	77 (1.7%)	5 (2.8%)	0.290	68 (2.2%)	5 (3.1%)	0.440
	Chronic Blood Loss	46 (1.0%)	3 (1.7%)	0.440	48 (1.5%)	2 (1.2%)	0.999
	Congestive Heart Failure	726 (16.0%)	42 (23.2%)	0.011	804 (25.6%)	37 (22.7%)	0.410
	Chronic Pulmonary Disease	1723 (38.1%)	69 (38.1%)	0.990	1534 (48.8%)	73 (44.8%)	0.320
	Coagulopathy	170 (3.8%)	27 (14.9%)	<0.001	122 (3.9%)	19 (11.7%)	<0.001
	Depression	622 (13.7%)	19 (10.5%)	0.210	319 (10.1%)	14 (8.6%)	0.520
	Diabetes, Uncomplicated	1323 (29.2%)	62 (34.3%)	0.150	1128 (35.9%)	42 (25.8%)	0.009
	Diabetes, Chronic Complications	184 (4.1%)	6 (3.3%)	0.610	116 (3.7%)	6 (3.7%)	0.996
	Drug Abuse	150 (3.3%)	3 (1.7%)	0.290	14 (0.4%)	0 (0%)	0.999
	Hypertension	2259 (49.9%)	79 (43.6%)	0.100	1987 (63.2%)	85 (52.1%)	0.005
	Hypothyroidism	563 (12.4%)	23 (12.7%)	0.920	636 (20.2%)	26 (16.0%)	0.180
	Liver Disease	136 (3.0%)	15 (8.3%)	<0.001	42 (1.3%)	4 (2.5%)	0.290
	Lymphoma	18 (0.4%)	1 (0.6%)	0.530	27 (0.9%)	1 (0.6%)	0.999
	Fluid/Electrolyte Disorders	1054 (23.3%)	75 (41.4%)	<0.001	875 (27.8%)	76 (46.6%)	<0.001
	Metastatic Cancer	225 (5.0%)	23 (12.7%)	<0.001	164 (5.2%)	12 (7.4%)	0.230
	Other Neurological Disorders	784 (17.3%)	28 (15.5%)	0.520	463 (14.7%)	24 (14.7%)	0.999
	Obesity	974 (21.5%)	47 (26.0%)	0.160	357 (11.4%)	12 (7.4%)	0.120
	Paralysis	663 (14.7%)	25 (13.8%)	0.750	246 (7.8%)	9 (5.5%)	0.280
	Peripheral Vascular Disorders	114 (2.5%)	4 (2.2%)	0.790	200 (6.4%)	14 (8.6%)	0.260
	Psychoses	231 (5.1%)	6 (3.3%)	0.280	91 (2.9%)	7 (4.3%)	0.300
	Pulmonary Circulation Disorders	223 (4.9%)	9 (5.0%)	0.980	146 (4.6%)	11 (6.7%)	0.220
	Renal Failure	525 (11.6%)	37 (20.4%)	<0.001	523 (16.6%)	41 (25.2%)	0.005
	Solid Tumor	473 (10.5%)	22 (12.2%)	0.470	370 (11.8%)	23 (14.1%)	0.370
	Peptic Ulcer	0 (0%)	0 (0%)		0 (0%)	0 (0%)	
	Valvular Disease	116 (2.6%)	2 (1.1%)	0.330	181 (5.8%)	16 (9.8%)	0.033
	Weight Loss	381 (8.4%)	28 (15.5%)	0.001	335 (10.7%)	25 (15.3%)	0.060
Tracheostomy Complication	Unspecified Complication	46 (1.0%)	0 (0%)	<0.001	26 (0.8%)	5 (3.1%)	0.001
	Infection	545 (12.0%)	12 (6.6%)		270 (8.6%)	14 (8.6%)	
	Mechanical Compilation	1539 (34.0%)	36 (19.9%)		975 (31.0%)	33 (20.2%)	
	Other Tracheostomy Complication	2394 (52.9%)	133 (73.5%)		1874 (59.6%)	111 (68.1%)	
Invasive Diagnostic Procedure		1715 (37.9%)	61 (33.7%)	0.250	1141 (36.3%)	64 (39.3%)	0.440
Surgical Procedure		1312 (29.0%)	35 (19.3%)	0.005	733 (23.3%)	30 (18.4%)	0.150
		**Mean (SD)**	**Mean (SD)**	** *p* **	**Mean (SD)**	**Mean (SD)**	** *p* **
Age, Years		49.62 (12.01)	53.36 (9.21)	<0.001	74.30 (6.93)	75.32 (6.76)	0.038
Time to Invasive Diagnostic Procedure, Days		1.76 (3.24)	2.93 (5.50)	0.990	1.74 (3.54)	4.02 (8.62)	0.051
Time to First Surgical Procedure, Days		2.54 (4.33)	3.63 (5.09)	0.260	2.71 (4.04)	5.14 (9.95)	0.440
Hospital Length of Stay, Days		6.48 (9.38)	10.10 (21.76)	0.035	6.32 (8.04)	11.18 (14.55)	0.014
Total Charges, Dollars		51,089 (81,336)	99,849 (127,214)	<0.001	50.548 (83.517)	100,726 (156,467)	<0.001

**Table 4 ijerph-19-09031-t004:** Characteristics of emergently admitted patients with the primary diagnosis of tracheostomy complications. Data were stratified according to surgery status, NIS 2005–2014.

Patients’ Characteristics		Adult (18–64), N (%)			Elderly (65+), N (%)		
		No Surgery	Surgery	*p*	No Surgery	Surgery	*p*
All Cases		3363 (71.4%)	1348 (28.6%)		2551 (77.0%)	764 (23.0%)	
Gender, Female		1474 (43.8%)	621 (46.1%)	0.160	1187 (46.5%)	391 (51.2%)	0.024
Race	White	1555 (53.9%)	637 (55.2%)	0.550	1436 (65.6%)	443 (66.9%)	0.290
	Black	838 (29.0%)	313 (27.1%)		429 (19.6%)	109 (16.5%)	
	Hispanic	319 (11.1%)	128 (11.1%)		182 (8.3%)	65 (9.8%)	
	Asian/Pacific Islander	74 (2.6%)	30 (2.6%)		63 (2.9%)	15 (2.3%)	
	Native American	11 (0.4%)	9 (0.8%)		8 (0.4%)	4 (0.6%)	
	Other	88 (3.1%)	36 (3.1%)		71 (3.2%)	26 (3.9%)	
Income Quartile	Quartile 1	1230 (37.6%)	487 (37.3%)	0.910	729 (29.3%)	225 (30.2%)	0.940
	Quartile 2	799 (24.4%)	322 (24.6%)		621 (24.9%)	184 (24.7%)	
	Quartile 3	738 (22.6%)	287 (22.0%)		609 (24.4%)	183 (24.6%)	
	Quartile 4	503 (15.4%)	211 (16.1%)		533 (21.4%)	153 (20.5%)	
Insurance	Private Insurance	772 (23.0%)	338 (25.1%)	0.025	198 (7.8%)	59 (7.7%)	0.480
	Medicare	1156 (34.4%)	402 (29.9%)		2204 (86.5%)	666 (87.3%)	
	Medicaid	1212 (36.1%)	496 (36.8%)		90 (3.5%)	24 (3.1%)	
	Self-Pay	99 (2.9%)	53 (3.9%)		17 (0.7%)	5 (0.7%)	
	No Charge	13 (0.4%)	4 (0.3%)		0 (0%)	1 (0.1%)	
	Other	106 (3.2%)	53 (3.9%)		38 (1.5%)	8 (1.0%)	
Hospital Location	Rural	219 (6.5%)	53 (3.9%)	<0.001	141 (5.5%)	26 (3.4%)	0.049
	Urban: Non-Teaching	1055 (31.4%)	302 (22.4%)		873 (34.2%)	257 (33.6%)	
	Urban: Teaching	2089 (62.1%)	993 (73.7%)		1537 (60.3%)	481 (63.0%)	
Comorbidities	AIDS	29 (0.9%)	6 (0.4%)	0.130	0 (0%)	0 (0%)	
	Alcohol Abuse	146 (4.3%)	49 (3.6%)	0.270	54 (2.1%)	13 (1.7%)	0.470
	Deficiency Anemias	772 (23.0%)	267 (19.8%)	0.018	744 (29.2%)	204 (26.7%)	0.190
	Rheumatoid Arthritis	54 (1.6%)	28 (2.1%)	0.260	55 (2.2%)	20 (2.6%)	0.450
	Chronic Blood Loss	40 (1.2%)	9 (0.7%)	0.110	31 (1.2%)	19 (2.5%)	0.011
	Congestive Heart Failure	556 (16.5%)	214 (15.9%)	0.580	649 (25.4%)	194 (25.4%)	0.980
	Chronic Pulmonary Disease	1292 (38.4%)	502 (37.2%)	0.450	1244 (48.8%)	367 (48.0%)	0.720
	Coagulopathy	135 (4.0%)	62 (4.6%)	0.370	108 (4.2%)	33 (4.3%)	0.920
	Depression	463 (13.8%)	179 (13.3%)	0.660	255 (10.0%)	78 (10.2%)	0.860
	Diabetes, Uncomplicated	975 (29.0%)	412 (30.6%)	0.290	870 (34.1%)	302 (39.5%)	0.006
	Diabetes, Chronic Complications	143 (4.3%)	47 (3.5%)	0.230	93 (3.6%)	29 (3.8%)	0.850
	Drug Abuse	109 (3.2%)	44 (3.3%)	0.970	12 (0.5%)	2 (0.3%)	0.440
	Hypertension	1661 (49.4%)	679 (50.4%)	0.540	1591 (62.4%)	487 (63.7%)	0.490
	Hypothyroidism	437 (13.0%)	150 (11.1%)	0.080	515 (20.2%)	148 (19.4%)	0.620
	Liver Disease	112 (3.3%)	39 (2.9%)	0.440	34 (1.3%)	12 (1.6%)	0.620
	Lymphoma	15 (0.4%)	4 (0.3%)	0.470	21 (0.8%)	7 (0.9%)	0.810
	Fluid/Electrolyte Disorders	814 (24.2%)	317 (23.5%)	0.620	748 (29.3%)	206 (27.0%)	0.210
	Metastatic Cancer	197 (5.9%)	52 (3.9%)	0.006	146 (5.7%)	31 (4.1%)	0.070
	Other Neurological Disorders	616 (18.3%)	196 (14.5%)	0.002	384 (15.1%)	104 (13.6%)	0.320
	Obesity	714 (21.2%)	308 (22.8%)	0.220	264 (10.3%)	105 (13.7%)	0.009
	Paralysis	507 (15.1%)	182 (13.5%)	0.170	201 (7.9%)	55 (7.2%)	0.540
	Peripheral Vascular Disorders	81 (2.4%)	37 (2.7%)	0.500	170 (6.7%)	47 (6.2%)	0.620
	Psychoses	172 (5.1%)	65 (4.8%)	0.680	79 (3.1%)	20 (2.6%)	0.500
	Pulmonary Circulation Disorders	162 (4.8%)	70 (5.2%)	0.590	128 (5.0%)	30 (3.9%)	0.210
	Renal Failure	402 (12.0%)	161 (11.9%)	0.990	450 (17.6%)	116 (15.2%)	0.110
	Solid Tumor	402 (12.0%)	94 (7.0%)	<0.001	329 (12.9%)	64 (8.4%)	0.001
	Peptic Ulcer	0 (0%)	0 (0%)		0 (0%)	0 (0%)	
	Valvular Disease	78 (2.3%)	40 (3.0%)	0.200	154 (6.0%)	43 (5.6%)	0.680
	Weight Loss	275 (8.2%)	134 (9.9%)	0.052	259 (10.2%)	101 (13.2%)	0.017
Tracheostomy Complication	Unspecified Complication	36 (1.1%)	10 (0.7%)	<0.001	29 (1.1%)	3 (0.4%)	<0.001
	Infection	514 (15.3%)	44 (3.3%)		263 (10.3%)	23 (3.0%)	
	Mechanical Compilation	851 (25.3%)	725 (53.8%)		647 (25.4%)	362 (47.4%)	
	Other Tracheostomy Complication	1962 (58.3%)	569 (42.2%)		1612 (63.2%)	376 (49.2%)	
Respiratory System Invasive Diagnostic Procedure		979 (29.1%)	714 (53.0%)	<0.001	742 (29.1%)	383 (50.1%)	<0.001
Digestive System Invasive Diagnostic Procedure		60 (1.8%)	89 (6.6%)	<0.001	71 (2.8%)	77 (10.1%)	<0.001
Deceased		146 (4.3%)	35 (2.6%)	0.005	133 (5.2%)	30 (3.9%)	0.150
		**Mean (SD)**	**Mean (SD)**	** *p* **	**Mean (SD)**	**Mean (SD)**	** *p* **
Age, Years		50.12 (11.77)	48.88 (12.27)	0.001	74.48 (6.97)	73.97 (6.76)	0.090
Time to Invasive Diagnostic Procedure, Days		1.47 (2.82)	2.25 (3.90)	<0.001	1.54 (3.07)	2.55 (5.44)	<0.001
Hospital Length of Stay, Days		5.46 (7.70)	9.51 (14.17)	<0.001	5.81 (7.70)	9.07 (10.49)	<0.001
Total Charges, Dollars		42,032 (64,680)	80,147 (114,965)	<0.001	43,983 (72,933)	82,925 (124,474)	<0.001

**Table 5 ijerph-19-09031-t005:** Multivariable backward logistic regression analysis to evaluate the associations between mortality and different risk factors in patients emergently admitted with a primary diagnosis of tracheostomy complications (NIS 2005–2014). Mortality was the dependent variable.

Patients’ Characteristics	Mortality	
	N = 9306	R^2^ = 0.142
	OR (95% CI)	*p*
Number of Events	N = 364	
Age, Years	1.007 (1.001, 1.013)	0.02
Hospital Length of Stay, Days	1.008 (1.001, 1.016)	0.018
Bacterial Infections (Other than Tuberculosis)	1.44 (1.12, 1.85)	0.004
Cardiac Diseases	3.21 (2.48, 4.15)	<0.001
Liver Diseases	2.61 (1.73, 3.93)	<0.001
Genitourinary System Diseases	1.39 (1.10, 1.76)	0.006
Fluid and Electrolyte Disorders	1.55 (1.24, 1.95)	<0.001
Neoplasms	1.63 (1.26, 2.10)	<0.001
Neurological Diseases	1.94 (1.54, 2.44)	<0.001
Platelet and White Blood Cell Diseases	1.52 (1.11, 2.10)	0.01
Trauma, Burns, and Poisons	1.90 (1.50, 2.42)	<0.001
Gender, Female		
Invasive Procedure	Removed Via Stepwise Backward Elimination	
Surgical Procedure		
Tracheostomy Complication Type		
Respiratory Diseases		
Coagulopathy		
Peripheral Vascular Diseases		
Cerebrovascular Diseases		
Tuberculosis		
Nonbacterial Infections		
Anemia and/or Hemorrhage		
Digestive Diseases other than Liver		
Diabetes		
Drug Abuse/Withdrawal/Dependence		
Alcohol Abuse/Withdrawal/Dependence		
Tobacco Use		
Hypertension		
Endocrine Diseases		
Nutritional/Weight Disorders		
Musculoskeletal System and Connective Tissue Diseases		
Psychiatric Diseases		
Skin Diseases		
Long Term Medication Usage		
Diseases of Oral Cavity, Salivary Glands, and Jaw		
Sleep Disorders		
Lack of Physical Evidence		
Inappropriate Diet and Eating Habits		
High Risk Lifestyle Behaviors		
Social Factors		

**Table 6 ijerph-19-09031-t006:** Lifestyle, comorbidities, and secondary diagnoses of patients emergently admitted with a primary diagnosis of tracheostomy complications (NIS 2005–2014). Data were stratified according to survival status.

	Adult, N (%)			Elderly, N (%)		
Lifestyle, Comorbidities and Secondary Diagnoses (ICD-9 Codes)	Survived	Deceased	*p* Value	Survived	Deceased	*p* Value
Observations	4524 (96)	181 (4)		3145 (95)	163 (5)	
Tuberculosis (010.0–018.96)	1 (0.0)	0 (0)	0.840	1 (0.0)	0 (0)	0.820
Bacterial Infections Other than Tuberculosis (020.0–041.9, 790.7)	806 (18)	67 (37)	<0.001	609 (19)	68 (42)	<0.001
Nonbacterial Infections (042, 795.71, V08, 045.0–139.8, 790.8, and/or presence of Comorbidity of AIDS)	433 (10)	9 (5)	0.038	198 (6)	15 (9)	0.140
Diabetes (250.0–250.93, V58.67, and/or presence of Comorbidity of Diabetes Uncomplicated or Diabetes Chronic Complications)	1513 (33)	71 (39)	0.110	1255 (40)	49 (30)	0.012
Hypertension (401.0–405.99, 796.2, and/or presence of Comorbidity of Hypertension)	2262 (50)	79 (44)	0.090	1987 (63)	85 (52)	0.005
Anemia and/or Hemorrhage (280.0–285.9, 784.7, 784.8, and/or presence of Comorbidity of Anemia)	1219 (27)	57 (32)	0.180	1109 (35)	70 (43)	0.046
Respiratory Diseases (415.0–417.9, 460–519.9, 784.91, 786, and/or presence of Comorbidity of COPD, ILD or Pulmonary Circulation Disease)	3604 (80)	162 (90)	0.001	2665 (85)	150 (92)	0.011
Coagulopathy (286.0–286.9, 790.92, V58.61, V58.63, and/or presence of Comorbidity of Coagulopathy)	452 (10)	30 (17)	0.004	361 (12)	28 (17)	0.028
Cardiac Diseases (391.X, 392.0, 393.398.99, 410.0–414.9, 420.0–429.9, 794.3X, 785.XX, and/or presence of Comorbidity of CHF or valvular diseases)	1705 (38)	126 (70)	<0.001	1867 (59)	129 (79)	<0.001
Cerebrovascular Diseases (325, 430–438)	335 (7)	17 (9)	0.320	323 (10)	13 (8)	0.340
Peripheral Vascular Diseases (440–457.9, and/or presence of Comorbidity of Peripheral Vascular Disorders)	282 (6)	18 (10)	0.045	317 (10)	23 (14)	0.100
Liver Diseases (570–573.9, 790.4, 794.8, and/or presence of Comorbidity of Liver Diseases)	168 (4)	20 (11)	<0.001	66 (2)	11 (7)	<0.001
Diseases of Digestive System other than Liver (530.00–569.9, 574.0–579.9, 787, 001.0–009.3, and/or presence of Comorbidity of Peptic Ulcer)	1494 (33)	57 (32)	0.670	1152 (37)	72 (44)	0.052
Diseases of Oral Cavity, Salivary Glands, and Jaws (520–529)	51 (1)	2 (1)	0.980	23 (1)	1 (1)	0.860
Nutritional/Weight Disorders (260–273.9, 275.XX,277.0–278.8, 783.XX, 799.3–799.4, and/or presence of Comorbidity of Weight Loss)	1911 (42)	84 (46)	0.270	1352 (43)	64 (39)	0.350
Endocrine Diseases (240.0–259.9, 991.0–992.9, and/or presence of Comorbidity of Endocrine Diseases)	1970 (44)	84 (46)	0.450	1659 (53)	71 (44)	0.022
Genitourinary System Diseases (580.0–629.9, 403.XX, 791.XX, 788.XX, and/or presence of Comorbidity of Renal Diseases)	1264 (28)	89 (49)	<0.001	1203 (38)	94 (58)	<0.001
Neurological Diseases (317.0–326, 330.0–337.9, 340–359.9, 392, 780.0–780.09, 780.2–780.4, 317–319, 290.XX, 294.XX, 781.0–782.0, and/or presence of Comorbidity of Paralysis or Other Neurological Disorders or Paralysis)	1699 (38)	109 (60)	<0.001	1053 (34)	74 (45)	0.002
Diseases of the Musculoskeletal System and Connective Tissue (274.XX, 710.0–739, and/or presence of Comorbidity of Rheumatoid Arthritis or Lupus)	680 (15)	22 (12)	0.290	524 (17)	24 (15)	0.520
Fluid and Electrolyte Disorders (275.0–276.9, 458.0–459.9, and/or presence of Comorbidity of Fluid and Electrolyte Disorders)	1325 (29)	88 (49)	<0.001	1001 (32)	87 (53)	<0.001
Neoplasms (140.0–239.9, V10.XX, and/or presence of Comorbidity of Lymphoma, Metastatic Diseases, or Tumor)	1235 (27)	58 (32)	0.160	1143 (36)	52 (32)	0.250
Platelet and White Blood Cell Diseases (204.0–208.92, 287.0–288.9, 238.71)	311 (7)	31 (17)	<0.001	249 (8)	21 (13)	0.024
Psychiatric Diseases (293.XX, 295.0–302.9, 306.0–316, 780.1, V62.8, V15.4, and/or presence of Comorbidity of Psychoses)	1187 (26)	33 (18)	0.016	624 (20)	30 (18)	0.650
Skin Diseases (680.0–709.9, 782.1–782.9)	961 (21)	44 (24)	0.320	655 (21)	36 (22)	0.700
Trauma, Burns and Poisoning (800–999)	937 (21)	69 (38)	<0.001	613 (20)	80 (49)	<0.001
Drug Abuse/Withdrawal/Dependence (292.0–292.9, 304.0–304.93, 305.2–305.93, and/or presence of Comorbidity of Drug Abuse)	154 (3)	3 (2)	0.200	16 (1)	0 (0)	0.360
Alcohol Abuse/Withdrawal/Dependence (291.0–291.9, 303.0–303.93, 305.0–305.03, and/or presence of Comorbidity of Alcohol Abuse)	189 (4)	6 (3)	0.570	66 (2)	1 (1)	0.190
Tobacco Use (305.1)	1003 (22)	27 (15)	0.021	585 (19)	21 (13)	0.070
Long-Term Medications/Radiotherapy (V58.0–V58-2, V58.62, V58.64–V58.66, V58.68–V58.69)	343 (8)	13 (7)	0.840	244 (8)	7 (4)	0.100
Social Factors (V60.0–V62.6, V63.0–V64.3, V15.81)	183 (4)	1 (1)	0.017	62 (2)	1 (1)	0.220
Sleep Disorders (327, 780.5, V69.4, V69.5)	852 (19)	23 (13)	0.038	346 (11)	7 (4)	0.007
Lack of Physical Exercise (V69.0)	0 (0)	0 (0)		0 (0)	0 (0)	
Inappropriate Diet and Eating Habits (V69.1)	0 (0)	0 (0)		0 (0)	0 (0)	
High Risk Lifestyle Behaviors (V69.2, V69.3)	0 (0)	0 (0)		0 (0)	0 (0)	
Body Mass Index of Less than 18.9 (V85.0)	73 (16)	1 (4)	0.300	38 (18)	3 (23)	0.360
Body Mass Index of 19–24.9 (V85.1)	27 (6)	2 (9)		23 (11)	3 (23)	
Body Mass Index of 25.0–29.9 (V85.21–V85.25)	20 (4)	0 (0)		20 (9)	0 (0)	
Body Mass Index of 30.0 and over (V85.30–V85.45)	345 (74)	20 (87)		134 (62)	7 (54)	

## Data Availability

Data are available upon request for verification purposes.

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
