# Peer review of "Mortality Risk Factors in Patients Admitted with the Primary Diagnosis of Tracheostomy Complications: An Analysis of 8026 Patients"

_ijerph, 2022, doi:10.3390/ijerph19159031_

Round 1

Reviewer 1 Report

Dear Editor,

I have read with great interest the manuscript proposed by Levy et al. There are some concerns that need to be clarified.

Introduction: the introduction appears without a clear logical thread, with subparagraphs being non clearly related to each other. I believe to revise especially the part related to the use of AI and how personal information may drive decision-taking. Also, the final sentence (i.e., "aims") should be introduced more properly, from the introduction is to broad and not clearly related to the reported aims.

Methods: the authors report the use of different statistical programmes and different chosen values p-values  (p < 0.01; p< 0.05) depending on the part. Also, p-value is two-tailed? Any test to study normal distribution of data? Data are presented (depending on the part) with M +/- SD or Mean (quartiles). The authors should clarify this issue. Also, the use of "sex" should be changed using "gender".

Author Response

Thank you so much for taking the time to provide thoughtful feedback on our article. We appreciate your comments and the opportunity to improve our work.

"Introduction: the introduction appears without a clear logical thread, with subparagraphs being non clearly related to each other. I believe to revise especially the part related to the use of AI and how personal information may drive decision-taking. Also, the final sentence (i.e., "aims") should be introduced more properly, from the introduction is to broad and not clearly related to the reported aims."

We worked on redirecting the introduction, with your specific points in mind. We hope now that it is more clear that we included the part about AI to strengthen our argument that a deeper dive into the risk factors of mortality for patients with a tracheostomy is important for AI to evaluate high-risk patients. 

Methods: the authors report the use of different statistical programmes and different chosen values p-values (p < 0.01; p< 0.05) depending on the part. Also, p-value is two-tailed? Any test to study normal distribution of data? Data are presented (depending on the part) with M +/- SD or Mean (quartiles). The authors should clarify this issue. Also, the use of "sex" should be changed using "gender".

We appreciate your comments. All mentions of "sex" are changed to "gender", per your suggestion.

Regarding the P values, we set the value of P<0.05 to be significant, however in our data analysis, some parts were even more significant and we wanted to represent that with the smaller P-value <0.01. Furthermore, we used a chi-square test for Tables 2-4 and as such, there is no standard deviation since the information is categorical and not continuous. Continuous variables are presented as mean (SD).

Reviewer 2 Report

I do not have any major concerns. However, it is unclear on what time period post procedure mortality refers. It seems very low for the complication of mechanical ventilation that is associated with high rates of complications itself. Thus this must be clearly stated inside the methods and throughout the paper as it probably refers only to immidiate hospitalization without acessing clinical course in referral ward or insitution. 

Author Response

Dear Reviewer,
Thank you so much for taking the time to provide thoughtful feedback on our article. We appreciate your comments and the opportunity to improve our work. 

We will be sure to include in the paper that post-procedure mortality is in reference to the immediate hospitalization. 
We thank you again for your contribution to this piece. 

Reviewer 3 Report

This is a good work but needs format / table editing.

It would better to remove Tables 3-6 to supplement and replace the included results with OR and 95% CIs. Accordingly, present them in forest plots would also add to the article.

In Table 2 remove the column of p-value- only in text, should this information be presented.

Author Response

Dear Reviewer,

Thank you so much for taking the time to provide thoughtful feedback on our article. We appreciate your comments and the opportunity to improve our work.

We have removed the p-value column from Table 2, per your suggestion.

Regarding your other suggestion, we believe that if we move these tables to the supplement then the message of the paper will be lost, and so we respectfully dissent.

Also, as this is not a systematic review, we elected not to use forest plots.
We thank you again for your contribution to this piece. 

Round 2

Reviewer 1 Report

Dear Editor,

the authors have edited the manuscript according to reviewer's concerns.

The manuscript is now ready for publication.